# New experimental diagnostics in combustion of forest fuels: Microscale appreciation for a Macroscale approach

Cancellieri Dominique[1], Leroy-Cancellieri Valérie[1*], Silvani Xavier[1], Morandini Frédéric[1].

[1]SPE - UMR CNRS 6134, University of Corsica, Corte, France

*Correspondence to*: Valérie Leroy-Cancellieri (vcancellieri@univ-corse.fr)

**Abstract.** In modelling the wildfire behaviour, a good knowledge of the mechanisms and the kinetic parameters controlling the thermal decomposition of forest fuel is of great importance. The kinetic modelling is based on the mass loss rate which defines the mass source term of combustible gases that supply the flames and influences the propagation of wildland fires. In this work, we investigated the thermal degradation of three different fuels using a multi-scale approach.

Lab-scale experimental diagnostics as Thermogravimetric Analysis (TGA), Differential Scanning Calorimetry (DSC), Cone Calorimeter (CC) or Fire Propagation Apparatus (FPA) led to valuable results for modelling the thermal degradation of vegetal fuels and allowed several upgrades of pyrolysis models.

But, these works remain beyond large-scale conditions of a wildland or forest fire. In an effort to elaborate the kinetic models under realistic natural fire conditions, a mass-loss device specifically designed for the field scale has been developed. The

15 paper presents primary results gained using this new device, during large-scale experiments of controlled fires. The mass-loss records obtained at field scale highlight the influence of the chemical composition and the structure of plants. Indeed, two species with similar chemical and morphological characteristics exhibit a similar mass-loss rate whereas the third present a different thermal behaviour.

The experimental data collected at the field scale lead to a new insight about thermal degradation processes of natural fuel,

when compared to the kinetic laws established in TGA. These new results, provide a global description of the kinetics of degradation of Mediterranean forest fuels. The results led to a proposed thermal degradation mechanism that has also been validated at a larger scale.

## 1 Introduction

In studying the forest fire propagation, kinetic modelling of thermal degradation mechanisms is one of the main prerequisites

for the determination of source terms, allowing for the development of realistic models. Numerous Computational Fluid Dynamics (CFD) codes have been developed for predicting the fire spread, the heat release and providing operational tools for the land managers (Linn et al., 2002; Mell et al., 2007). Indeed, physically-based models, initiated by Grishin (Grishin, 1997), account for each mechanism of heat transfer individually and predicts not only the spread rate of the fire but also its complete behavior. The thermal degradation of the solid phase as well as the combustion of the gaseous pyrolysis products are described,

requiring the development of specific kinetic models for the vegetation fuels. Appropriate kinetic mechanisms should be coupled with the description of transport mechanisms (heat-, mass-, and momentum-transfer) to provide a more detailed process simulation. The mass-loss rate of the solid is one of the most important parameters in describing the evolution of the solid phase. Indeed, it is directly linked to the mass loss rate due to pyrolysis and represents the initial factor of the combustion process. Such parameters are often determined from small-scale tests such as ThermoGravimetric Analysis (TGA).

TGA is a thermoanalytical technique commonly used in solid-phase thermal degradation studies (Ninan, 1989; White et al., 2011). It has gained widespread attention in the thermal analysis of biomass pyrolysis (Di Blasi, 2008; White et al., 2011). TGA measures a decrease in the substrate mass caused by the release of volatiles (devolatilisation) during the thermal decomposition. In practice, the mass of the sample being heated at a specific rate is monitored as a function of temperature or time. TGA requires sufficiently small samples for the diffusion effects to be negligible and for the pyrolysis process to be kinetically controlled (Miller and Bellan, 1997). The experimental data collected under perfectly controlled TGA conditions ensures an accurate determination of the kinetic mechanism. Unfortunately, these experimental conditions are not realistic in term of heating rate with those encountered in forest fire. Sometimes, calorimetric experiments are performed with a Cone Calorimeter (CC) (Schemel et al., 2008) or a Fire Propagation Apparatus (FPA) (Simeoni et al., 2012), but the gap to the real scale is still significant.

Overall, field data collection is demanding and potentially dangerous; however, it is considered the best alternative for improving and validating the fire spread models (Morandini et al., 2006). Consequently, a number of field tools have been proposed; such as thermocouples, heat flux gauges (Silvani et al., 2009), gas sensors (Miranda et al., 2010) audio and video sensors (Stavrakakis et al., 2014).

In order to preserve the plant structure some authors have carried out tests on trees (Mell et al., 2009) or litter (Dupuy, 1995), but the ignition conditions are not similar to those encountered during a wildfire. However, according to our knowledge, the accuracy measurement of mass loss have never be done in field experiment conditions.

In view of these limitations, the aim of this study is to propose kinetic models adapted to realistic Mediterranean forest fire conditions. A mass-loss device specifically designed for the field scale has been developed for this purpose. This device can record the mass-loss and temperatures of three vegetation samples submitted to a heat flux from a spreading flame front across a large bed of fuel. One of the main advantage of this system is to submit simultaneously 3 different samples to the same fire front and in identical meteorological conditions, which greatly facilitates the comparison of the thermal behaviours. The choice of this heat flux source allowed to achieve a better correspondence to forest fire conditions. This system can provide characteristic dynamic data such as temperature, mass-loss rate.

In a first time reactional mechanisms are defined from experiments performed in perfectly controlled conditions on thermal thin samples (in TGA). Using these experimental data, kinetic models are proposed for each species. Thanks to these models, the simulation of the mass loss rate is done at higher heating rates, the same as the one measured during field experiment. In the last time, the mass loss rates obtained from the simulation are compared to the experimental data collected at field scale.

Very few fire studies are performed directly in the open field focusing on the on-line measurement and monitoring of fuel mass bring about open fires.

## 2 Materials

### 2.1 Samples

The forest fuels used were representative of Mediterranean land, we have selected 3 species with different physiological structures: Rockrose (*Cistus Monspeliensis*), Heather (*Erica Arborea*), Pine (*Pinus Pinaster*). The forest fuels were collected from live shrubs or tree in neighboring forests, close to the University of Corsica. The experiments were performed using branches and twigs with still attached leaves or needles. Some authors have demonstrated that only small particles (< 6mm) are considered in governing the dynamic of fire spread (Burrows, 2001; Morvan and Dupuy, 2004). According to this observation we have decided to sample the foliage and aerial parts of each species. So the proportion of leaves and twigs vary for each specie with a close ratio, around 50% of leaves, for heather and rockrose due to their similar structure. Conversely, the pine is mainly composed by needles for around 75% compared to twigs. In order to highlight the different structure of specie the figure 1 presents a picture of each specie.

Figure 1

In order to focus on oxidative pyrolysis and combustion processes, samples were oven-dried for 24h at 333K (Leroy-Cancellieri et al., 2014). This sample state allows to suppress the dehydration phenomenon and thus the influence of the moisture content on the burning. Moreover, the fact of getting rid of the moisture content let us concentrate on the influence of the physicochemical parameters of plants on the burning rate. So collected samples were then brought to the laboratory, washed with deionized water and oven-dried for 12 hours at 333K. After these preparation stages, the sampling has been separated in 2 cases:

- For field experiments, the aim was to keep conditions encountered during of wildand fire, so we have used an intact branch of dried plant in order to be close as possible to their natural state. For each specie, only one branch is directly placed on the prototype tube. The initial mass of the samples is approximatively $50 \pm 0.001$g which is larger by a factor of around 10000 than that of the samples for the experiments performed at the laboratory scale. According to the species it represents a branch about 20cm high (*cf.* figure 1).

- For TGA experiments, dried samples were grounded and sieved to pass through a 100μm mesh, then kept to the desiccator to protect them from ambient humidity. The sieved powdery sample was stored in airtight plastic containers for future use.
The moisture content arising from self-rehydration was about 4% for all the samples before testing.

The chemical and physiological properties play a significant role on thermal decomposition of fuel, so a characterization of the studied species including elemental and lignocellulosic composition, and physiologic properties have been performed. Lignocellulosic materials were determined by different gravimetric methods, according to normalized (Ona et al., 1994; Tappi, 1974) or published methods (Peterssen, 1984; Wise et al., 1946). The density was measured following the methodology proposed by Moro (Moro, 2006). The elemental analysis was carried out at the SCA (Service Central d'Analyse) USR 59 CNRS, and the results are shown in Table 1.

Table 1

## 2.2 Laboratory experiments

Mass loss is one of the main parameters used for the kinetic characterisation of thermal degradation mechanisms. It is known as the major driving parameter for the characterisation of source terms. Moreover, the mass loss provides qualitative and quantitative data on different reactions which take place in the heated solid (Kissinger H.E., 1957). In order to investigate the mass loss bahavior, thermogravimetric experiments were carried out in a thermogravimetric analyzer (PerkinElmer, Pyris 1 TGA). For each sample, 5mg of dried above-ground biomass was heated from 350K to 900K under dynamic conditions at a heating rate of $30K.min^{-1}$. The TGA furnace was flushed with air at a rate of $20mL.min^{-1}$ to maintain the oxidative atmosphere for thermal degradation of particles in the course of experiments. Each experiment was repeated at least thrice with an excellent reproducibility higher than 99.7%. The precision of temperature measurements was ±2K.

## 2.3 Field experiments

To investigate the scale effect and to highlight the similarities and differences between laboratory and field experiments, a device especially designed for field has been created. It allows the simultaneously records of the mass-loss and the temperature when samples are exposed to a heat source. In order to achieve the real fire conditions, the heat source is a fire front. The description of this mass-loss prototype and its usage is detailed in the following sections.

### 2.3.1 Differential mass loss prototype setup

The prototype consists of two parts: the one responsible for measurements and the one responsible for data acquisition. The characteristics of each part are given below. Figure 2 depicts the entire mass-loss prototype.

Figure 2

Analog measurement devices

The device was sized to be one-fifth the width of the plot to burn. This ensures that the fire completely encompasses the system during its propagation.

The device includes three load cells integrated in a welded ceramic box (1260mm × 170mm × 100mm), covered with a 50mm thick refractory lining (Thermal Ceramics Kaowool 1600).

Taking into account the fact that the meteorological and fire conditions are difficult to reproduce perfectly, we decided to install the three load cells on the apparatus to follow the behaviour of three species subjected to the same fire propagation. With such three species available in the prototype, the differential analysis between the samples can be performed independent of the external conditions.

The three load cells (LSB 200, Futek$^®$) have a maximum capacity of 450g ± 0.1 % and having a width of 6.8mm, a height of 19mm, and a length of 17.5mm. Samples are introduced in the load cells through the stainless tube mounted on top of each cell. The height and the diameter of this tube (190mm × 20mm) were reckoned up to avoid any lift effect. Moreover, the position and the height of the sample in the tube can be adjusted to optimize the interaction between the flame and the sample. The distance between each supporting tube (500mm) was chosen such that the decomposition of a particular branch could not affect the neighbouring branch. Thus, the plant will only be affected by the fire front in front of the device.

To measure the temperature acting on the sample and the heating rate of the fire, another tube, accommodating a thermocouple, is place very close to the tube supporting the branch. The thermocouple, positioned in a vertical position can be adjust in height. With the possibility of an adjustment of each element (branch and thermocouple), we ensured that the thermocouple is positioned at mid-height of the branch. This configuration ensures the determination of the real temperature to which the sample is subjected.

The K-type thermocouples were selected according to their temperature range, with an upper limit of 1300 ± 0.5°C (Omega$^®$ HKMTSS-010G-8, diameter: 25μm).

Data acquisition and process

The acquisition system is integrated into a thermal box with a Multi-Layer Aluminization (Z-Flex®) shield. A remote wireless acquisition system is actually impossible to use because of the disturbances introduced by the thermal shield. The temperature inside the thermal box is controlled using a thermocouple. If necessary, it is adjusted by a fan, when the temperature of the thermal box is rising. A laptop located inside the box is used to transmit the data simultaneously through a USB interface using custom software.

The mass-loss data are recorded using the Sensit software of Futek with a frequency of 2.5Hz.

The temperature data are synchronised with the mass-loss data recorded with the same frequency, by the acquisition unit Omega® TC-08. This system can accommodate up to eight thermocouples with an acquisition frequency of 10Hz.

One of the main advantages of this prototype is that three different species can be subjected to the same external heating conditions in line and be analysed simultaneously under the same field conditions.

### 2.3.2 Experimental and meteorological conditions

The field tests took place in an open field terrain with no slope, situated in the Unit Instruction and Civil Security Intervention No. 5 of Corte in Corsica. The three species are subjected to the same heat source: the fire spreading over a wood–wool bed. This fuel was selected for the reasons of good repeatability of the heating conditions.

About 120 kg of weight wood-wool was used, forming a bed of 10 m at length and almost 6 m at width. The average height of the bed has been selected appropriately to comply with a fuel load of 2 kg.m$^{-2}$. The orientation of the fuel bed was based on the meteorological forecast of the day of the experiment, following the wind direction. A linear ignition was performed at the bottom of the wood wool bed.

The wind velocity and direction were recorded using a two-dimensional ultrasonic anemometer at 2.5m above the ground surface to reflect the average wind acting on the fire front. The anemometer was located in the direction of the propagation (at the end of the plot). The wind data were recorded using another (synchronised) data logger at a sampling rate of 1Hz. The average velocity of the wind measured during the experiment was 1.2 m.s$^{-1}$ and its direction was close to South Est (143°). During the campaign the wind was relatively constant with a standard deviation of 0.44 m.s$^{-1}$.

Heat flux measurement

Three types of thermal transfers occur during the spread of a natural fire. Because the paper focuses on the thermal degradation governed by a heat source external to the sample, the heat transfers inside the flame front were not considered in the present work. Beyond that, it is first important to explain here why we chose to focus on radiation and convection. Heat conduction in a solid fuel is usually not considered in the set of transport phenomena involved with the spread of a natural fire as detailed in (Silvani et al., 2012).

Furthermore, this previous work on thermal transfers from a fire to a bed of pine excelsior were already performed in similar no slope, no wind conditions and with comparable fuel loads (Silvani et al., 2012).
This study exhibit that the heat transfer is mainly due the thermal radiation from the fire. At such scales, the vegetal fuel is also known to absorb the heat radiation according coefficients to black body conditions (Boulet et al., 2011).

The measurements of the heat fluxes emitted from the flame front during the fire spread are therefore measured using radiant heat flux gauge from Medtherm. It consists in a total heat flux gauge upon which a special window is used in order to eliminate the convective heat transferred to the sensing area. As in the present case, the use of a sapphire window was especially appropriate for detecting the radiative properties of a natural fire at the bench scale. Spectral emission experiments (Boulet et al., 2011) conducted on flames and their related burning bed of excelsior and vine branches illustrate that, first, the main radiation proceeds from the flame and, second, flames and fuel beds mainly emit in the wavenumber ranges of $2000-2300$ cm$^{-1}$ and $1500-6000$ cm$^{-1}$, respectively. The sapphire window [1.2 μm; 5.5 μm] was therefore convenient for the present study because it transmitted thermal radiation emitted in the wavenumber range of $1800-8333$ cm$^{-1}$. The radiant heat flux emitted from the flame front during fire spread were measured using a transducer located at 1 m from the end of fuel bed. The measurement height was 0.5 m. The heat flux measurements were recorded using a data logger, at a sampling rate of 1 Hz. The transducer was calibrated by the manufacturer in the range of $0-200$ kW.m$^{-2}$ and had a response time lower than 0.25 s. Because the radiant gauge was calibrated using its sapphire window (with a 3% error for the incident heat flux in the range $0-150$ kW.m$^{-2}$ and 10% beyond this according to the manufacturer), no transmittance correction was needed for the heat flux density. Furthermore, as presented in the same reference, one cannot omit the possible contribution of faster fluctuations occurring at a larger scale than the present one, due to the turbulent nature of the flow. At the end, one must accept that investigating the scale dependence of spectral properties in the heat transfers via radiation from natural fires remains an open valuable subject.

Under these limitations, the radiant heat flux gauge provided accurate measurements of the longitudinal component of the heat flux density when they faced an approaching fire front. The transducer was oriented toward the flame front and had a 150° viewing angle. The region viewed by the gauge, located at 1 from the litter's end, allowed to measure the contribution of the whole fire front. Finally, a thermocouple recorded the body temperature of the gauge during fire tests, for controlling it worked in nominal conditions.

The devices deployed on the site are depicted in the figure 3 which presents the overall experimental setup.

Figure 3

The position of the prototype within the plot was tested in different configurations: the device was firstly placed in the middle of the field but the steady state conditions of the fire propagation were not obtained. Then the position at the completely end of the plot was tried but we observed an edge effect due to the lack of litter behind the prototype. Figure 4 shows pictures of the device during these two tests configuration.

Figure 4

Finally, the prototype has been placed near the end of the fuel bed to ensure the steady state propagation without edge effect. The experiments were performed in the area with no slope and were replicated three times on the same day to ensure identical surrounding conditions

## 3 Results

### 3.1 Laboratory experiments

Using TGA, the thermal degradation in air is characterised by a continuous weight loss until the point when the weight becomes almost constant. The first derivative of such thermogravimetric curves (i.e. $-dm/dt$) yields the maximum reaction rate. Such a procedure is known as Derivative Thermogravimetry (DTG). The character of the TGA curve, in combination with the corresponding DTG peaks, gives a clear indication of the number of stages in thermal degradation.

Figure 5 presents the experimental results on the thermal degradation of fuels heated at $30K.min^{-1}$ from 350 to 900K under air sweeping.

Figure 5

A clear similarity in the decomposition process can be observed for the three species. Such similarities imply that there could be a general kinetic scheme to describe biomass thermochemical degradation under air atmosphere. TGA curves exhibit two stages of weight loss, which are confirmed in DTG by two peaks (*cf*. Figure 5). Despite the complex chemical process, experimental data suggest that a two-step model of global reactions can describe the most important features of the thermal and oxidative degradation of plants. The first mass loss due to decomposition begins slowly and accelerates rapidly in the temperature range of 500-550K. The second mass loss follows the first one and reaches an overall mass loss of more than 90%. Moreover, the oxidative process is claimed to have two stages. The first stage is the volatilisation of main biomass compounds and the production of char residue at low temperatures. The second stage includes the decomposition of lignin and the combustion of the charcoal produced at the preceding stage (Fang et al., 2006). The same phenomena were observed and recorded by other authors as well (Branca and Di Blasi, 2004; Safi et al., 2004; Shen et al., 2009).

In order to compare biomass thermal behavior, DTG is frequently used to determine several temperature indexes: ignition temperature (Ti), final temperature of the first process ($Tf_1$), final temperature of the second process ($Tf_2$). Ti is defined as the inflection point of DTG at start of the degradation, while $Tf_{1,2}$ are defined as the inflection points at the end of each stage.

Table 2

For the heating rate considered in this study, the onset temperature is the lowest for Heather, higher for Pine, and yet higher for Rockrose. This observation can be used as the ignition criterion, since the onset temperature marks out the beginning of oxidation reactions. The fuels with low onset temperatures are most ignitable, and they burn easily. These results will be compared to the field-scale experiments.

Using the TGA data, a kinetic model for each specie will be done in the section 4.

### 3.2 Field experiments

Figure 6 presents the average temperature as a function of time, according to the data recorded while the plot was burning.

Figure 6

The temperature profiles were nearly the same for the three species with a maximum at 926K. The behaviour of the temperature indicates that the effect of the fire front on the prototype can be considered as a straight line. Using the evolution of temperature during the course of the experiment, the heating rate was estimated for each plant. The average of the heating rates during the heating phase were obtained: 13.2K.s$^{-1}$ for Pine, 12.9K.s$^{-1}$ for Heather, and 12.1K.s$^{-1}$ for Rockrose. The heating rate can be approximate at a median value of 12.7±0.6 K.s$^{-1}$ for all species. This result accredits the fact that the temperature measurement in 3 points with 500 mm difference is homogenous along the whole test field.

Figure 7 demonstrates the mass loss synchronised to the temperature *vs.* time data shown above. The figure exhibits, for each specie, the records of the 3 experiments and their average, with a confidence level lower than 16%.

Figure 7

In order to facilitate the comparison between the fuels, the figure 8 shows on the same graph the average mass loss.

Figure 8

Heather starts to lose its mass more quickly than Rockrose and Pine. This observation is in agreement with the laboratory experiments and ignition temperature provided in table 2. Indeed, Heather exhibits a low onset temperature implying that this species will ignite prior to the two other species. The behaviour exhibited in TGA is similar as the one observed at the field scale. The order of the rate of degradation among species determined at the laboratory scale is kept at the field scale (the degradation is fast for Heather and Rockrose, and it is slower for Pine).

The main advantage to the field experiments is the exposure of samples to a real fire front so it is of importance to have the measures of heat fluxes records during the experimental tests. The average longitudinal distribution of radiant heat flux impinging ahead of the flame front is provided in Figure 9.

Figure 9

The preheating of the fuel via radiation is a long range process. The main advantage of such fire experiments in the field is to allow the samples of vegetative fuel to be exposed to thermal conditions characteristic of a spreading wildfire (Silvani et al. 2009). For the configuration considered in the present study, the rise of temperature of the fuel elements the can occur far from the fire front since sufficient radiation levels ($>5$ kW.m$^{-2}$) are measured at a distance of 6 m and pyrolysis can begin within 2 m from the fire front. Indeed, piloted and auto-ignition of such fuels was observed for laboratory scale experiments for impinging heat fluxes in the range of 20-25 kW.m$^{-2}$ (Bartoli et al., 2011). It should be noticed that, even if high radiation levels in the same order of magnitude can be obtained in the laboratory using radiant heaters, the heating rates are not representative of a travelling fire front since samples are submitted to a constant heat flux.

## 4 Kinetic analysis

Thermogravimetric data were used to find the best set of kinetic parameters for our three species. Using TGA measurements, the conversion degree α is defined as:

$$\alpha = \frac{m_0 - m}{m_0 - m_f} \tag{1}$$

With $m$ the mass, $m_0$ the initial mass and $m_f$ the final mass of the sample.

When the TGA experiments are conducted under non-isothermal conditions, the rate of heterogeneous solid-state reactions can be described as:

$$\frac{d\alpha}{dt} = \frac{1}{\beta} A e^{-\frac{E_a}{RT}} f(\alpha) \tag{2}$$

where $f(\alpha)$ is the conversion function (reaction model), $A$ is the pre-exponential factor, $E_a$ is the activation energy, $R$ is the universal gas constant, and $\beta$ is the heating rate.

The kinetic parameters $A$, $E_a$, and $f(\alpha)$ can be estimated from the TGA experimental data by a variety of techniques (Vyazovkin et al., 2011; Vyazovkin and Wight, 1998). To obtain a reliable kinetic description of the processes under investigation, we used an approach that combines accurate isoconversional methods with model-fitting methods (Chrissafis, 2009; Pratap et al., 2007). This two-step approach is the so-called Hybrid Kinetic Method (HKM), which was developed earlier by Cancellieri *et*

*al.* (Cancellieri et al., 2005). At the first step, isoconversional methods provide $E_a(\alpha)$ and the reaction model. At the second step, these initial data are used in a model-fitting technique to obtain the pre-exponential factor and the $n^{th}$-order model. Such a multi-step procedure allows the selection of the models that might otherwise be indistinguishable because of the poor quality of the regression fit when performed independently. In other words, such an approach gives the highest probability of selecting

5  the most accurate kinetic triplet ($A$, $E_a$, and the model).

The laboratory experiments for our three species support the following two-step kinetic mechanism. The first process is modelled as

$$Virgin_{(s)} \rightarrow Char_{(s)} + Gas_{(g)} \quad (I)$$

10  The second reaction deals with the oxidation of the chars produced during the first process:

$$Char_{(s)} \rightarrow Residue_{(s)} + Gas_{(g)} \quad (II)$$

Both reactions (I) and (II) can be described by the following differential equations (we consider an $n^{th}$-order model):

$$\frac{d\alpha_1}{dt} = \frac{1}{\beta} A_1\, e^{-\frac{E_{a1}}{RT}} (1 - \alpha_1)^{n_1} \tag{3}$$

$$\frac{d\alpha_2}{dt} = \frac{1}{\beta} A_2\, e^{-\frac{E_{a2}}{RT}} (\alpha_1 - \alpha_2)^{n_2} \tag{4}$$

The use of HKM allowed us to obtain the kinetic parameters listed in Table 3.

Table 3

**4.1 Comparisons**

Considering the two-stage mechanism and the kinetic parameters listed in Table 3, the mass-loss of each species can be simulated at the average heating rate measured during field experiment, which is 12.7±0.6 K.s$^{-1}$ for all species. These numerical simulations were then compared to the experimental mass-loss recorded at field-scale. Figure 10 compares the mass loss

25  obtained experimentally for each species and the data of modelling according to Eqs. (3) and (4).

Figure 10

Generally, simulations are in good agreement with the experimental mass-loss rate even if there are some differences (as an

30  attempt). For Rockrose, the model does not accurately fit the data in the range $0.85 > m/m_0 > 0.60$, probably because the initiation and preheating mechanisms are more complex than in the model described by the simple Arrhenius equation of order

n. For Heather, experiments exhibit an accelerated degradation process which can be explained by the very fine structure of this species with branches as short as 2mm. On the contrary, Pine consists of single branches of 6mm diameter. Such thickness of the sample is responsible for the incomplete degradation.

The species are characterized in order to provide suitable input parameters. With accurate input parameters, models implemented CFD codes could predict the propagation of fire. To obtain a reliable description of the processes and a good agreement of simulation modellers need to know the mass-loss rate on the whole degradation. This parameter $\dot{m}$ is defined as:

$$\dot{m} = \frac{(m_0 - m_f)}{(t_0 - t_f)} \tag{5}$$

With $m$ the mass, $t_0$ the initial time and $t_f$ the final time of the experiment.

To validate the kinetic mechanism ant its parameters, the mass-loss rate obtained from field scale experiments is compared to the one determined for the simulation performed with the kinetic model. These data are summarized in the table 4.

Table 4

The main scope of this large scale test campaign is to reveal the thermal behaviour of different species in the same experimental conditions. The data obtained in the table 4 highlight the importance to take into account the physiological and chemical nature of species. Indeed, the mass-loss rate of pine is 50% lower than the heather and 40% lower than the rockrose. The significant differences must be integrate in detailed physics-based models to ensure a reliable characterization of source terms.

With the aim to provide guidelines for an integration in detailed physics-based models a radar chart has been used to graphically determined which chemical or structural parameters is most impacting the mass-loss rate. For coherence reasons with the target element ($\dot{m}$) the graphic, presented in figure 11, is only focused on the parameters directly linked with the mass-loss: cellulose, lignin, holocellose and fuel density. For more visibility, all the parameters have been dimensionless.

Figure 11

The figure 11 reveals that the holocellulose is the main impacting parameter on the mass-loss rate. Conversely, extractive is inversely proportional to mass-loss rate. Usually, physics-based models take into account fuel density, well it seems that the chemical composition and the structure of the plants are of primary interest when modelling wildland fire.

**5 Conclusion**

The stochastic conditions of fire imply great difficulties for the reproducibility of measurements. For these conditions, a differential mass-loss prototype has been designed with the aim to validate kinetic models adapted to the field scale.

Comparatives mass losses data on three different plant species have never been recorded simultaneously. Moreover, it is the first time that the kinetics of decomposition of biomass have been validated under real wildland fire conditions, thus ensuring reliable characterization of source terms.

The technology presented in this paper is based on a completely new approach where the development of a new field mass loss device, combined with recent progress in the understanding of the behavior achieves never before recorded data.

An experimental device, perfectly adapted to the biomass specificity based on a completely new differential approach has been developed. The prototype has been tested with three Mediterranean species. The results collected from field experiments emphasise the influence of various parameters such as: the ignition temperature, biomass type and anatomical structure. However, the two-stage kinetic model based on the TGA data, seems to fit well the experimental data obtained at the field scale. Using the field scale measurements, the kinetic validity of the scheme is then extended outside TGA. This study allowed to validate the kinetics of decomposition of various biomasses under real wildland fire conditions, thus ensuring reliable characterization of source terms. However, there were flawed predictions caused by the natural physiology of the samples (thickness and size of the leaves and branches). In fact, the initiation stage of preheating is strongly related to the physiology. Further studies will be focussed on the integration of sample thickness in the model as an inhibiting parameter.

**Acknowledgement**

The authors thank the Unit Instruction and Civil Security Intervention No. 5 of Corte in Corsica for the provision of their experimental field.

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

Table 1: Fuel characterization.

| Specie / Parameter | Rockrose | Heather | Pine |
|---|---|---|---|
| Carbon (%) | 46.58 | 52.43 | 50.54 |
| Hydrogen (%) | 6.22 | 6.98 | 6.76 |
| Oxygen (%) | 37.68 | 35.92 | 41.53 |
| N, mineral matter (%) | 9.52 | 4.63 | 1.70 |
| Holocellulose (%) | 52 | 54.3 | 43.4 |
| Lignin (%) | 34.4 | 34.4 | 38.9 |
| Extractives (%) | 9.2 | 9.2 | 13.1 |
| Fuel density (kg.m$^{-3}$) | 960 | 544 | 511 |

Table 2: Characteristic temperatures of thermal degradation of species at $\beta = 30K.min^{-1}$

| Specie \ Parameters | $T_i$ (K) | $T_{f_1}$ (K) | $T_{f_2}$ (K) |
|---|---|---|---|
| Rockrose | 563 (1.2) | 648 (1.0) | 834 (1.4) |
| Heather | 538 (1.5) | 631 (1.3) | 827 (1.9) |
| Pine | 549 (0.8) | 643 (0.5) | 818 (1.1) |

Table 3: Summary of kinetics parameters (Cancellieri et al., 2013).

| | $Virgin_{(s)} \rightarrow Char_{(s)}+Gas_{(g)}$ | | | $Char_{(s)} \rightarrow Residue_{(s)}+Gas_{(g)}$ | | |
|---|---|---|---|---|---|---|
| | $n_1$ | $Ea_1$ (kJ.mol$^{-1}$) | $A_1$ | $n_2$ | $Ea_2$ (kJ.mol$^{-1}$) | $A_2$ |
| Rockrose | 3.74 | 120 | 22.10 | 0.52 | 128 | 14.50 |
| Heather | 2.63 | 80 | 12.90 | 0.52 | 114 | 12.50 |
| Pine | 3.97 | 118 | 23.20 | 0.43 | 128 | 14.20 |

Table 4: Mass-loss rates of each species

| | $\dot{m}_{exp}$ (g.s$^{-1}$) | $\dot{m}_{sim}$ (g.s$^{-1}$) | Relative error |
|---|---|---|---|
| Rockrose | 0.01508 (±11.6%) | 0.00941 | 0.37599 |
| Heather | 0.01842 (± 15.9%) | 0.01084 | 0.41151 |
| Pine | 0.00907 (± 12.8%) | 0.00851 | 0.06174 |

| | $\dot{m}_{exp}$ (g.s$^{-1}$) | $\dot{m}_{sim}$ (g.s$^{-1}$) | Relative error |
|---|---|---|---|
| Rockrose | 0.01508 (±11.6%) | 0.00941 | 0.37599 |
| Heather | 0.01842 (± 15.9%) | 0.01084 | 0.41151 |
| Pine | 0.00907 (± 12.8%) | 0.00851 | 0.06174 |

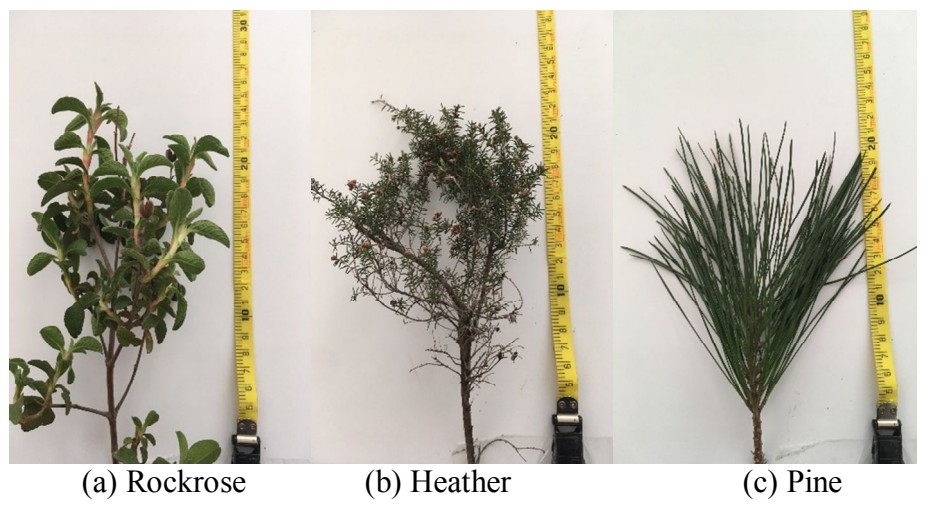

(a) Rockrose  (b) Heather  (c) Pine

Figure 1: Picture of samples

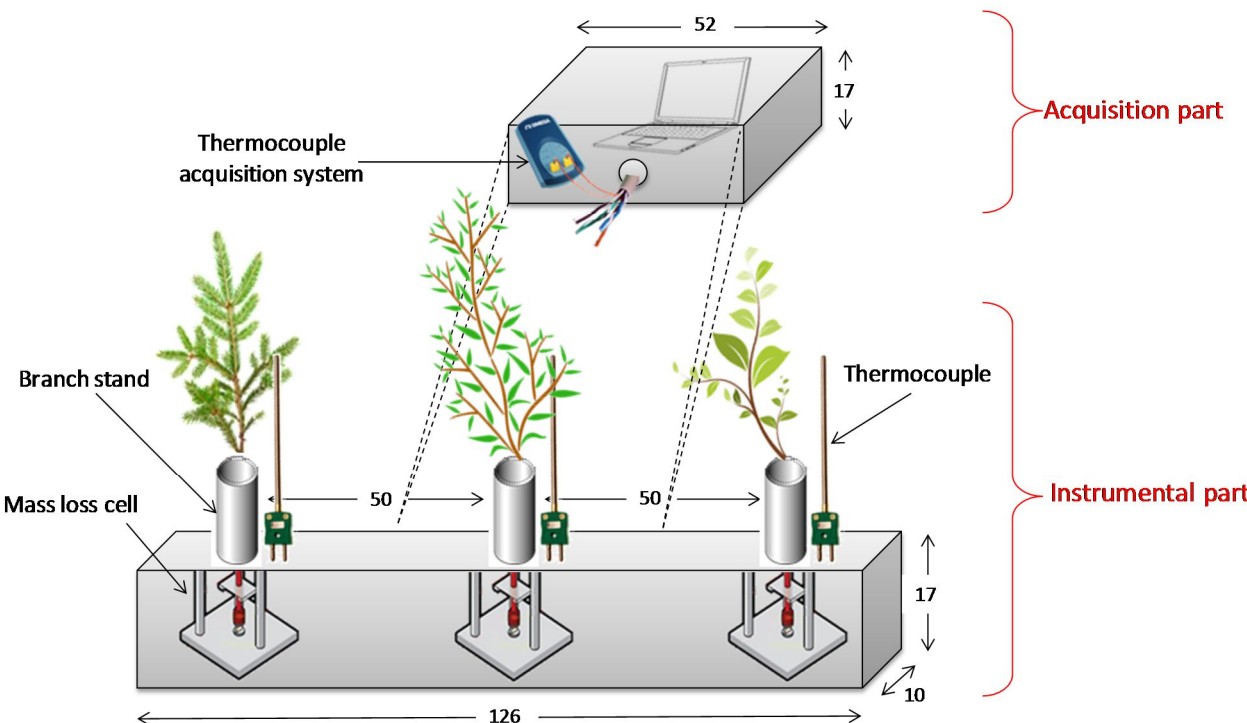

Figure 2: The differential mass-loss prototype (measurements are expressed in cm)

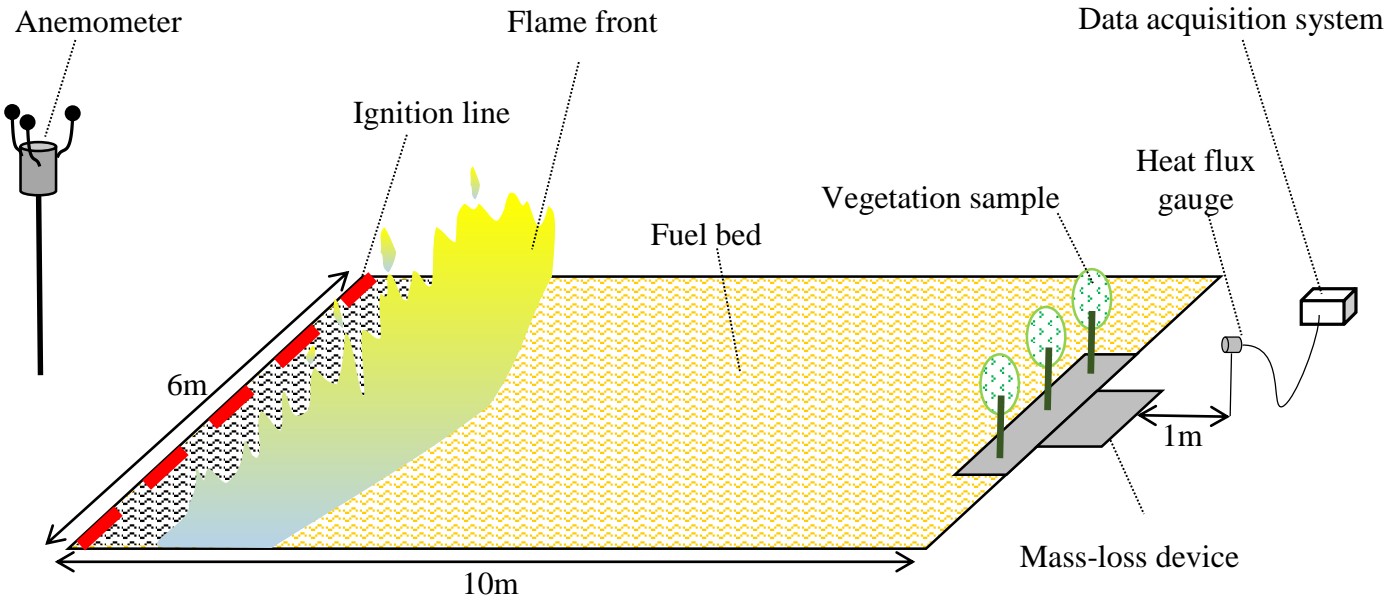

Figure 3: Experimental configuration

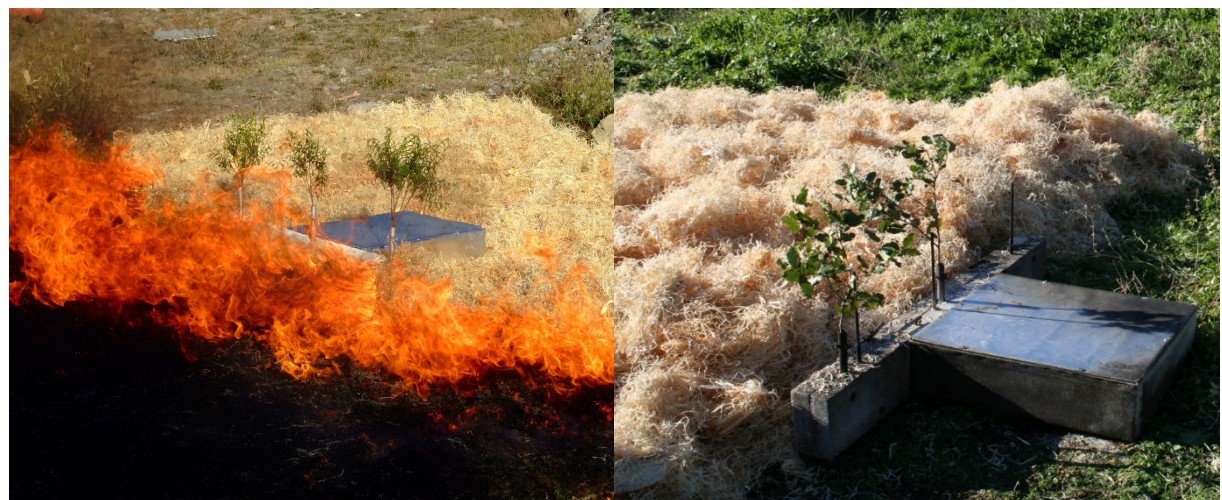

Figure 4: Differential mass loss prototype placed in the middle (left) and at the end (right) of the fuel bed.

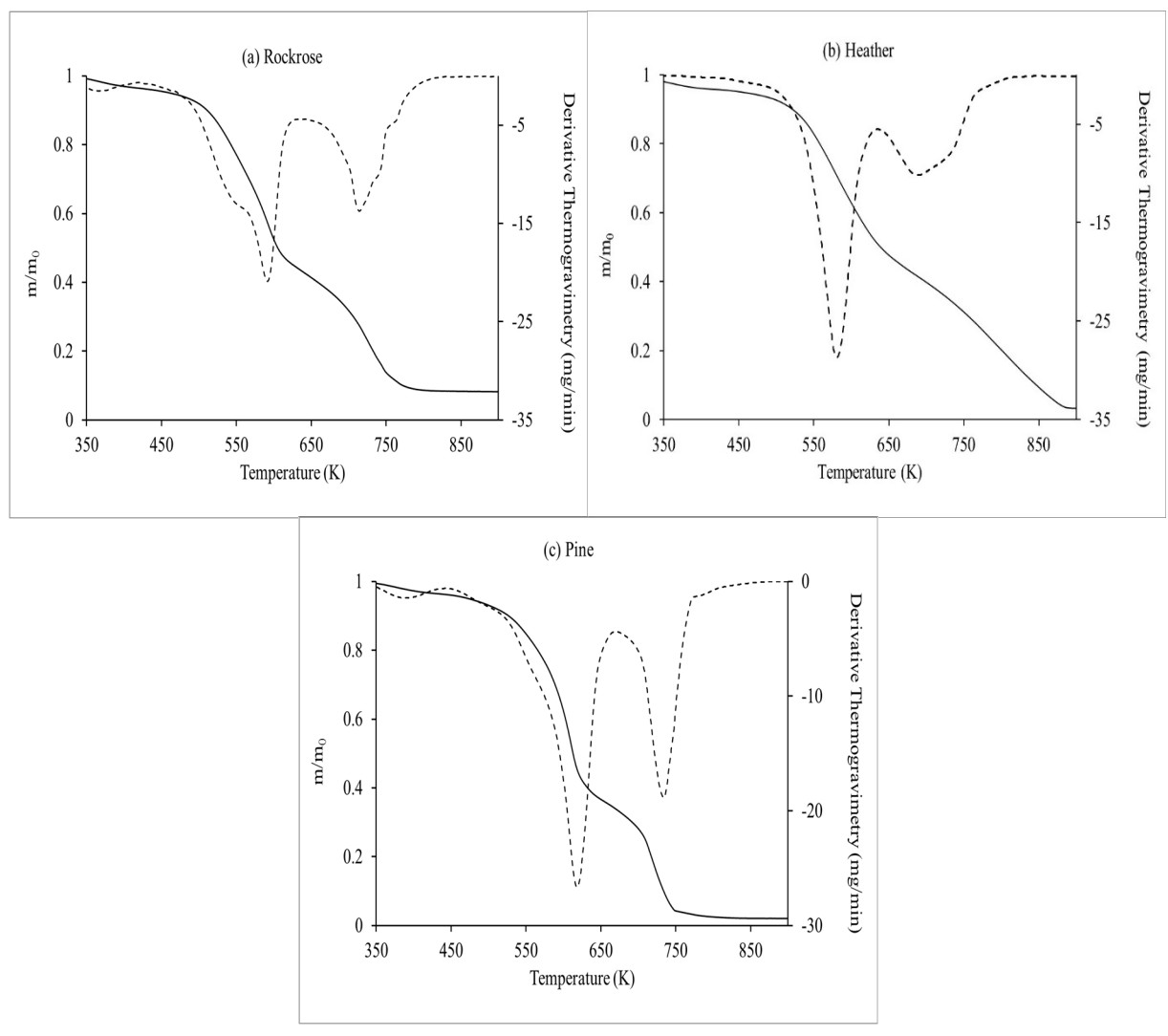

Figure 5: TGA (lines) and DTG (dotted) of oven dried of Rockrose (a), Heather (b), Pine (c) samples obtained with a linear heating rate of 30 K/min under air atmosphere.

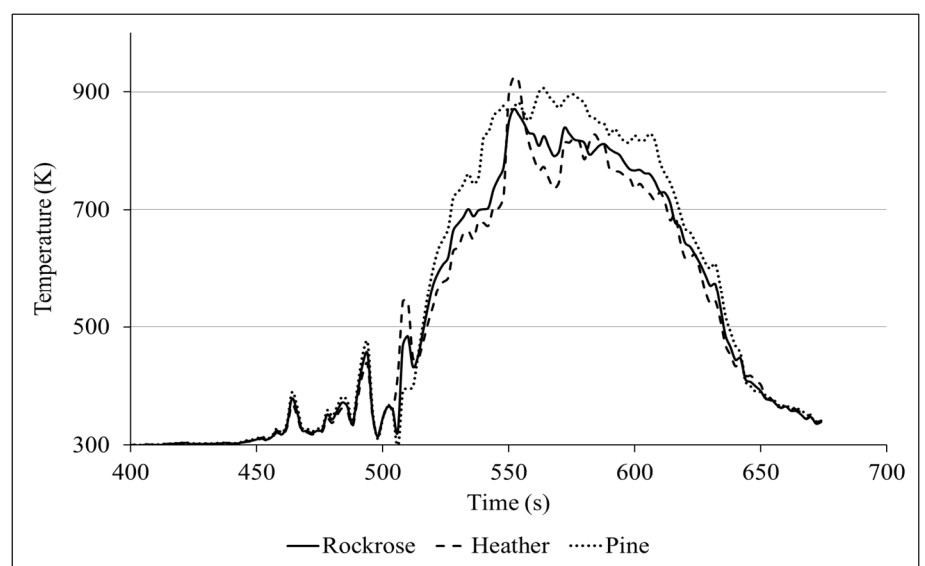

Figure 6: Average temperature *vs.* Time obtained during a field experiment.

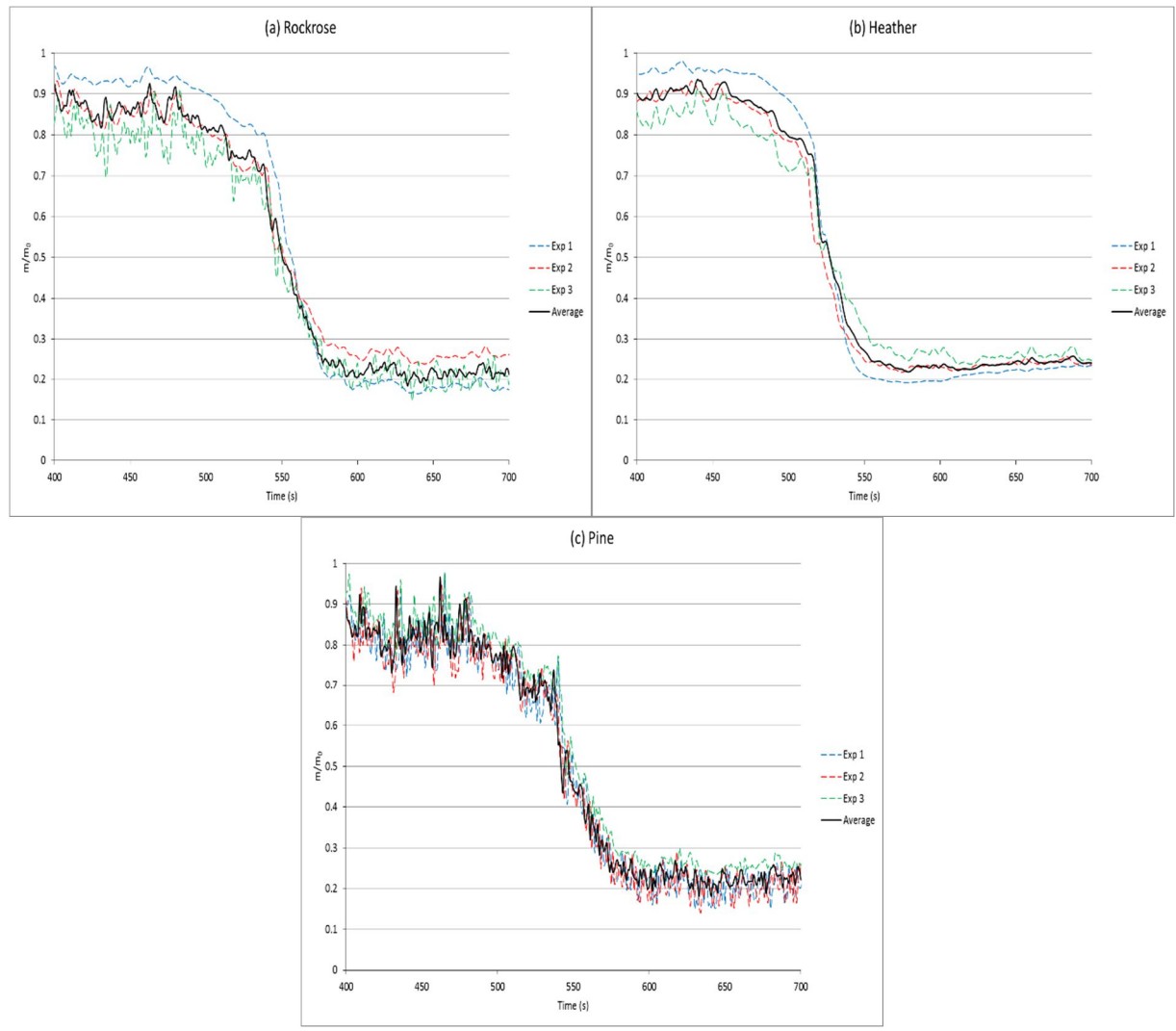

Figure 7: Experimental mass losses obtained at field scale.

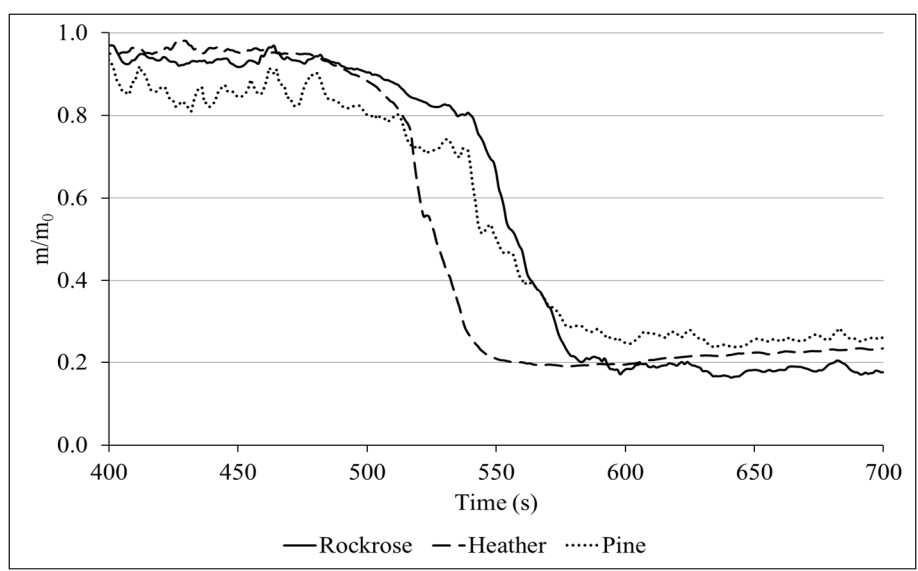

Figure 8: Comparison of experimental mass losses of the 3 species.

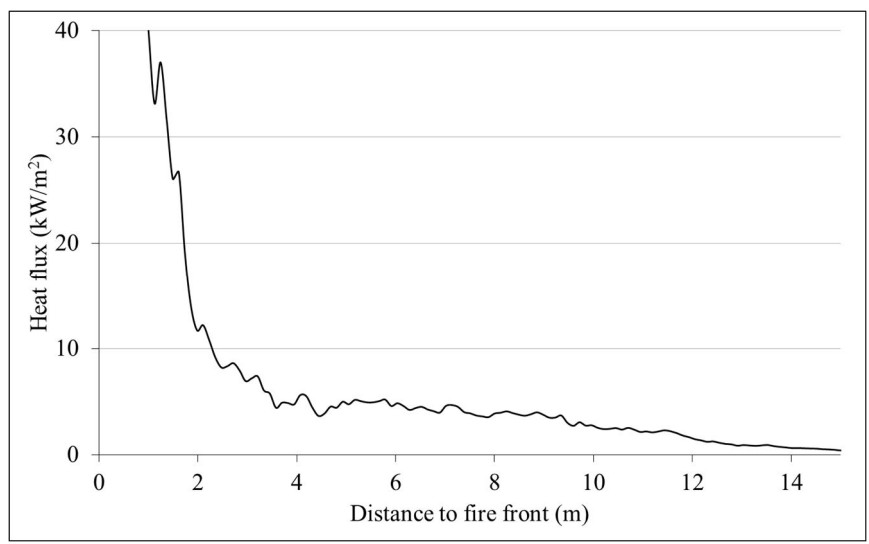

Figure 9: Average heat flux measurement according to the distance from the fire front.

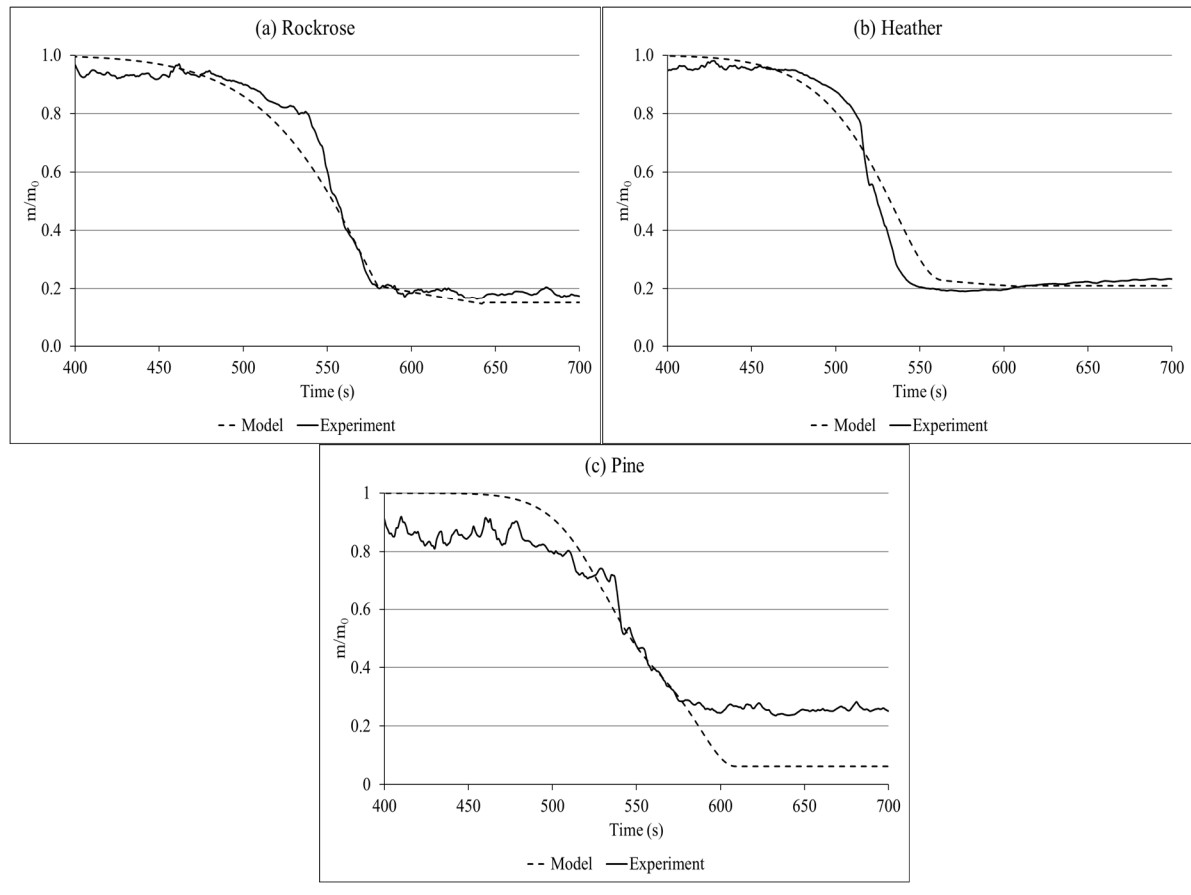

Figure 10: Experimental (solid line) and modeled (dashed line) mass loss for (a) Rockrose, (b) Heather and (c) Pine.

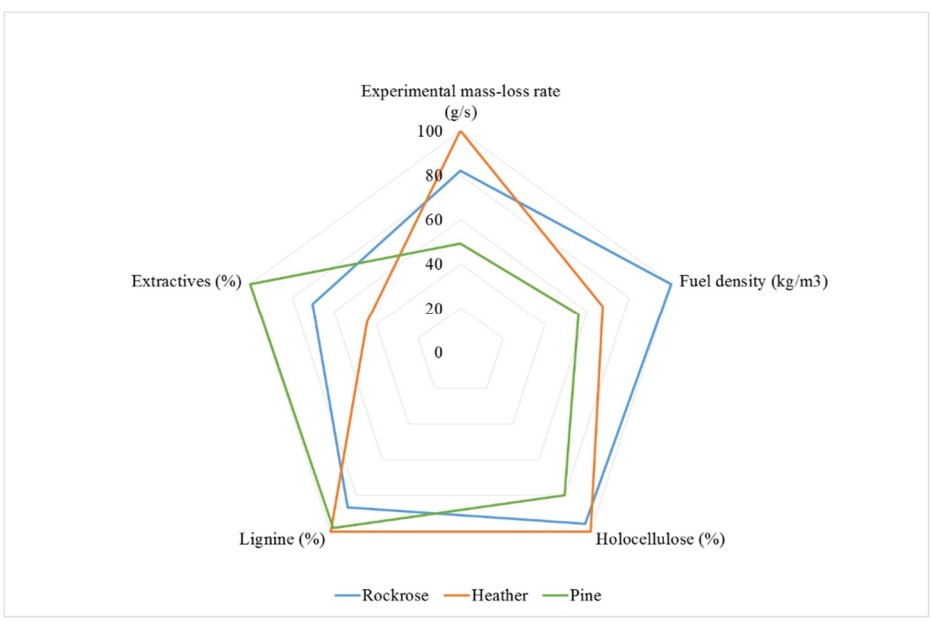

Figure 11 : Radar chart of the experimental mass-loss rate and the chemical and structural parameters.