# Peer review of "New experimental diagnostics in combustion of forest fuels: Microscale appreciation for a Macroscale approach"

_Natural Hazards and Earth System Sciences, 2017_

## Referee Comment (RC1) · Anonymous Referee #1 · 2 Mar 2018

This study attempt to relate between using parameters issued from small scale testing methods in very specific conditions and their application to describe the burning dynamics of large scale fires. The developed methodology is impeccable and the analysis is well balanced between showing the benefits of the proposed approach and its limitations due to the lack of heat transfer and physiology considerations.

Overall, I think that this study is very valuable for the community but it needs more elaboration, especially since it has already been published in 2014, with the exact figures, methodology and results, under the following: D. Cancellieri, V. Leroy-Cancellieri, E. Leoni, Multi-scale kinetic model for forest fuel degradation, in Advances

in Forest Fire Research, 2014. http://dx.doi.org/10.14195/978-989-26-0884-6_40 URL: http://hdl.handle.net/10316.2/34103

In order to avoid repetition of what is already published, I propose that the authors include novel elements, such as more in depth analysis on the differences between laboratory scale and field scale measurements, or perhaps testing the proposed kinetic model in a CFD code, as it is already suggested by the authors. Other elements can also be added, if they are novel. Only for that reason I consider that major revisions are necessary. Please consider my recommendation in a positive light.

More detailed comments on the manuscript are listed below:

Page 2 (line 8-10): Wind and heat flux conditions can be very similar in FPA and cone calorimeter than in real scale conditions, could you elaborate on how the gap between real scale and laboratory scale tests is significant?

Page 2 (line 15): It is worth mentioning that Dupuy in international journal of wildland fire (1995) measured the mass loss for an intermediate scale fire spread on a tilted table and Mell et al. measured the mass loss for a single Douglas fir tree in Combustion and Flame (2009)

Page 2 (line 28): The last sentence is incorrect as numerous publications by Morvan, Mell, Rochoux and others reported back on the use of kinetics models implemented in physical models and compared to field data.

Page 3 (2.1 samples): In the field experiments, were the samples living fuels? Is there an estimation of their fuel moisture content? It would be valuable to at least mention it, since, the evaporation process is not included in the model.

Page 5 (2.3.2 experimental and meteorological conditions): Was there any measurement of the flame height or of the heat flux received by the sample? This is important in order to relate to real scale and laboratory scale fire conditions. Page 5 (2.3.2 experimental and meteorological conditions): I understand the purpose of placing the

samples at the edge of the fuel bed. However, placing the samples in the middle of the fuel bed would have provided more realistic fire conditions, such as more radiation from the back of the flame front and more induced wind. This could have significant impact on the temperature and mass loss curves at t>550s. What there any other technical limitations for choosing this configuration?

Page 7 (Figure 5): Could you add more explanation on the significance of the first small peaks that are reached around 480-510sec? Do they represent local ignitions? Are these small peaks included in the "straight line" described line 7? Is this simplification overlooking the influence of evaporation process on the mass loss, especially for pine? Page 8 (line 18), There is a typographical error in the equation. Apostrophe to be removed

Page 9 (line 25): The authors highlight the importance of taking into account the physiological nature of species and to integrate them in CFD models. This is the exact same conclusion from the authors study in 2014. Could the author provide results or even guidelines on of the implementation of this model in a CFD code?

Page 9-10 (Conclusion): What it the conclusion on the similarities and the differences between laboratory and field experiments? Mass loss and temperature can be measured in laboratory as well, why aren't they compared?

---

## Referee Comment (RC2) · Anonymous Referee #2 · 4 Mar 2018

The paper is devoted to mass loss measurements of three natural fuels under laboratory (TGA, DSC) and field (custom made setup) conditions and subsequent calculation of thermokinetic parameters using an approach developed earlier. The authors raised a very important problem of scale effect on thermal decomposition of fuels; however, I did not find any discussion of the problem and comparison of their results with the literature. I also find the description of the methods and presentation of the results to be somewhat insufficiently detailed and clear. The English language requires some polishing as well. Nevertheless, the results are interesting and have a potential to be published in the peer reviewed journal. Therefore, I recommend a major revision. Specific comments are given below.

[Figure]

The shape of the fuel sample, as well as its "biological structure", will significantly influence its thermal degradation. Two samples with different number of twigs will burn differently. What was the exact procedure of selecting samples for field experiments, what was the number of twigs and leaves? This also will effect TGA results. As the sample was 5 mg, did it contain only crushed leaves, or twigs as well? If yes, what was the proportion? All of this must have significant effect on the thermal degradation of samples. Full description of the sample preparation procedure should be added to Section 2. Also, after reading the paper it is not clear to me if only one field experiment or several were conducted as all the results are presented as a single measurement. There are no confidence levels and comparison of repetitions.

The authors should add more analysis and discussion to the results. Why wasn't the difference of MLR between the experiment and the simulation for pine significant compared to other species? Why MLR of pine two times slower than those of Rockrose and Heather? What is the difference between the obtained kinetic parameters and those found in the literature with regard to multi-scale approach?

The authors should highlight throughout the paper that the obtained results are applicable for surface fires. More intense fires will give different heating and mass loss rates and can result in a mismatch between experimental and simulation results. It would also be worthwhile describing limitations of their approach.

Additionally, I have the following minor comments: 1. The abstract needs to be rewritten. Specific results and conclusions should be added. 2. Section 2.1. A picture of samples needs to be added 3. It's better to move Fig.1 to the beginning of section 2.3.1 4. Page 4, lines 21-23. It is not clear where the thermocouples were located, at the end of the tube or mid-height of the fuel brunch? 5. There is no reference to Fig. 3 in the text. 6. Page 6, line 16. Temperature units need to be changed to K. 7. Figure 4. Axis ticks are needed on temperature axis. It is also hard to see them on other axes. 8. Figure 5, Table 1-3. It would be worthwhile adding confidence levels. 9. I would recommend merging Fig. 5 and 6. 10. Table 3. A column with relative or absolute

errors needs to be added.

---

## Author Comment (AC1) · 4 Apr 2018

(1) Overall, I think that this study is very valuable for the community but it needs more elaboration, especially since it has already been published in 2014, with the exact figures, methodology and results, under the following: D. Cancellieri, V. Leroy- Cancellieri, E. Leoni, Multi-scale kinetic model for forest fuel degradation, in Advances in Forest Fire Research, 2014. In order to avoid repetition of what is already published, I propose that the authors include novel elements, such as more in depth analysis on the differences between laboratory scale and field scale measurements, or perhaps testing the proposed kinetic model in a CFD code, as it is already suggested by the

authors. Other elements can also be added, if they are novel. Only for that reason I consider that major revisions are necessary. Please consider my recommendation in a positive light.

Reply : The work presented at the Conference of Coimbra was focused on prototype design. It aimed at presenting our approach and our intents regarding the measurement of mass loss in real fire conditions. The results published in the proceeding of the conference only came from a single experiment. Since we have tripled the experiments to provide reproducible data which have been included in the revised version of this paper. In order to take into account the reviewer comment, new elements, such as the chemical composition, the heat flux measurement and the confidence levels, have been added. It is well known that the chemical and physiological properties play a significant role on thermal decomposition of fuel, so we proposed in the revised paper a deeper characterization of studied species including elemental and lignocellulosic composition, and physiologic properties (table 1). Moreover, as underline by the reviewer in its comments (n°6), the heat flux is a great of importance in modelling wildland fire, so we have included a large part devoted to the heat flux measurement carry out during our tests campaign. These data, in part, will fill the lack of information on heat transfer and physiological parameters. All these elements are new and are discussed and correlated to the thermal behaviour of each species.

(2) Page 2 (line 8-10): Wind and heat flux conditions can be very similar in FPA and cone calorimeter than in real scale conditions, could you elaborate on how the gap between real scale and laboratory scale tests is significant?

Reply: For our point of view, FPA and cone calorimeter experiments are still far from conditions obtained at field scale. Indeed, the samples studied are not kept intact, since it must be placed in a basket about 12-13 cm in diameter by 3-5 cm high. On the other hand, during the laboratory experiments, the samples are suddenly submitted to a constant heat flux, while during field tests the samples are subjected to a fire front. The samples are heated by a real flame not by infrared heater unlike using FPA or with

a radiant panel when using a calorimeter cone. The whole heating conditions ensure the conditions of preheating and the variations of fluxes (with increasing fire intensity) encountered in real conditions.

(3) Page 2 (line 15): It is worth mentioning that Dupuy in international journal of wildland fire (1995) measured the mass loss for an intermediate scale fire spread on a tilted table and Mell et al. measured the mass loss for a single Douglas fir tree in Combustion and Flame (2009).

Reply: This oversight has been rectified.

(4) Page 2 (line 28): The last sentence is incorrect as numerous publications by Morvan, Mell, Rochoux and others reported back on the use of kinetics models implemented in physical models and compared to field data.

Reply: This sentence has been deleted.

(5) Page 3 (2.1 samples): In the field experiments, were the samples living fuels? Is there an estimation of their fuel moisture content? It would be valuable to at least mention it, since, the evaporation process is not included in the model.

Reply: In order to focus on oxidative pyrolysis and combustion processes, samples were oven-dried even at field scale. Because the fuel moisture content is more impacting the burning rate than the type of fuel, the fact of getting rid of the moisture content let us concentrate on the influence of the physicochemical parameters of the plants on the mass-loss rate. For more clarity, this part (2.1. samples) has been rewritten and more details on the sampling process have been done.

(6) Page 5 (2.3.2 experimental and meteorological conditions): Was there any measurement of the flame height or of the heat flux received by the sample? This is important in order to relate to real scale and laboratory scale fire conditions.

Reply: In order to characterize the heat flux impacting the sample, we have also deployed a flux meter on the experimental site (cf. figure 3). A part dedicated to the

description (cf. section 2.3.2) and to the results (cf section 3.2) of these measures has been added in the text.

(7) Page 5 (2.3.2 experimental and meteorological conditions): I understand the purpose of placing the samples at the edge of the fuel bed. However, placing the samples in the middle of the fuel bed would have provided more realistic fire conditions, such as more radiation from the back of the flame front and more induced wind. This could have significant impact on the temperature and mass loss curves at t>550s. What there any other technical limitations for choosing this configuration?

Reply: We have tested different configurations: in the middle of the fuel bed (as shown the picture of the figure 4) and at the end. The configuration where the prototype was positioned in the middle only allowed a 5m propagation which did not ensure the quasi-stationary conditions that we were looking for. It would have been interesting to work on a longer fuel bed and position the prototype at around 10m but the site of the UICS n°5 did not allow it.

(8) Page 7 (Figure 5): Could you add more explanation on the significance of the first small peaks that are reached around 480-510sec? Do they represent local ignitions? Are these small peaks included in the "straight line" described line 7? Is this simplification overlooking the influence of evaporation process on the mass loss, especially for pine? Reply: In order to have any disturbances of the thermocouple on the sample and in return, we have placed the thermocouple on the side of each sample. Moreover, we have selected K-type thermocouples (diameter: $25\mu$m) for their high sensibility and their temperature range, with an upper limit of $1300 \pm 0.5°$C. With this configuration and the high sensibility of thermocouples, we detect the fluctuations due to the eddy of the flames approaching the target samples. This is what is visualizing around 480-510 s.

(9) Page 8 (line 18), There is a typographical error in the equation. Apostrophe to be removed.

Reply: The apostrophe has been removed.

(10) Page 9 (line 25): The authors highlight the importance of taking into account the physiological nature of species and to integrate them in CFD models. This is the exact same conclusion from the authors study in 2014. Could the author provide results or even guidelines on of the implementation of this model in a CFD code?

Reply: We do agree with the reviewer that we have to provide guidelines for modelling mass-loss rate. With this aim, we have graphically determined which chemical or structural parameters is most impacting the mass-loss rate. For coherence reasons with the target element (m ÌĞ) we have focused on the parameters directly linked with the mass-loss: cellulose, lignin, holocellose and fuel density. For more visibility, we have dimensionless all the parameters. This graphic (figure 11) reveals that the holocellulose is the main impacting parameter on the mass-loss rate. Conversely, extractive is inversely proportional to mass-loss rate. Usually, CFD models take into account fuel density, well it seems that the chemical composition and the structure of the plants are of primary interest when modelling wildland fire.

(11) Page 9-10 (Conclusion): What it the conclusion on the similarities and the differences between laboratory and field experiments? Mass loss and temperature can be measured in laboratory as well, why aren't they compared?

Reply: In TGA, experiments are carried out under dynamic conditions, ie at a pre-set heating rate. We have worked at 30 K/min which represents the maximum heating rate for which there is no temperature gradient between the order and the measurement. Whereas in field experiments, we have measured heating rates upper than 10 K/s. So in order to compare mass-loss rate, we have used the kinetic model describe by the eq. 3 and 4, and simulated the mass-loss rate at the same heating rates that the one observed on field (i.e 12.7 K/s). For our point of view, it is the only way to compare mass-loss rates. Nevertheless, due to the stochastic and uncontrolled conditions of field experiment the confidence level are more important with 16% vs. 0.0.3% of error

for TGA. With regard to the measurement of heat flux, it should be noted that, even if the high levels of radiation are of the same order of magnitude as those obtained in the laboratory, the heating rates are not representative of an ambient fire front since the samples are subjected to a constant heat flux.

Please also note the supplement to this comment:
https://www.nat-hazards-earth-syst-sci-discuss.net/nhess-2017-451/nhess-2017-451-AC1-supplement.pdf
* * *
[Figure]

[Figure]

(a) Rockrose          (b) Heather          (c) Pine

Figure 1: Picture of samples

[Figure]

Figure 2: The differential mass-loss prototype

Acquisition part

Instrumental part

[Figure]

Figure 3: Experimental configuration

[revised manuscript text omitted]

---

## Author Comment (AC2) · 4 Apr 2018

1. Major comments

(1) The shape of the fuel sample, as well as its "biological structure", will significantly influence its thermal degradation. Two samples with different number of twigs will burn differently. What was the exact procedure of selecting samples for field experiments, what was the number of twigs and leaves? This also will effect TGA results. As the sample was 5 mg, did it contain only crushed leaves, or twigs as well? If yes, what was the proportion? All of this must have significant effect on the thermal degradation of samples. Full description of the sample preparation procedure should be added to

Section 2.

Reply: We do agree with the reviewer, this part of the manuscript need more details on the sampling process. Some authors have demonstrated that only small particles (< 6mm) are considered in governing the dynamic of fire spread (Burrows, 2001; Morvan and Dupuy, 2004). According to this observation we have decided to sample the foliage and aerial parts of each species. So the proportion of leaves and twigs vary for each specie with a close ratio, around 50% of leaves, for heather and rockrose due to their similar structure. Conversely, the pine is mainly composed by needles for around 75% compared to twigs. To highlight the different structure of specie a picture of each specie (fig. 1) has been inserted in the revised paper.

After the selection of samples, they were brought to the laboratory, washed with deionized water and oven-dried for 12 hours at 333K.

After these preparation stages, we can separate the sampling in 2 cases: - For field experiments, the aim was to keep conditions encountered during of wildand fire, so we have used an intact branch of dried plant. For each specie, only one branch is directly placed on the prototype tube.

- For TGA experiments, dried samples were grounded and sieved to pass through a 100$\mu$m mesh, then kept to the desiccator. The sieved powdery sample was stored in airtight plastic containers for future use.

All these informations have been added in the manuscript

(2) Also, after reading the paper it is not clear to me if only one field experiment or several were conducted as all the results are presented as a single measurement. There are no confidence levels and comparison of repetitions.

Reply: For each type of experiment (field and TGA), the tests have been repeated 3 times. For TGA, records exhibit an excellent reproducibility higher than 99.7%. For field experiment, the confidence level is lower than 16%. According to the reviewer

comments, a figure (fig. 7) presenting 3 field experiments and their average has been added into the revised manuscript.

(3) The authors should add more analysis and discussion to the results. Why wasn't the difference of MLR between the experiment and the simulation for pine significant compared to other species? Why MLR of pine two times slower than those of Rockrose and Heather? What is the difference between the obtained kinetic parameters and those found in the literature with regard to multi-scale approach?

Reply: 2 main parameters could be attributed to the different of MLR. First of all, as mentioned by the reviewer, the chemical composition of the species influence the thermal degradation. Secondly, this work highlight the important role of the structure of plant. Indeed, the pine exhibit a very different thermal behavior since it is mainly composed by needles. In order to focus on the influence of these parameters on the thermal behavior of plants, we have added a radar chart (fig. 11) correlating the mass-loss rate to chemical and structural parameters.

(4) The authors should highlight throughout the paper that the obtained results are applicable for surface fires. More intense fires will give different heating and mass loss rates and can result in a mismatch between experimental and simulation results. It would also be worthwhile escribing limitations of their approach.

Reply: In the field conditions experimented in this work, we think that the gap between TGA and field experiments is enough important. This work demonstrates that despite the various change of sample: mass (5 mg vs. 20g), structural form (pulverized vs. intact branch) and in experimental conditions especially the heating rates (30 K/min vs. 10 K/s) the kinetic model obtained at micro-scale can predict in a main way the thermal degradation of wildland fuel. This approach is only focused on thermal degradation of solid fuel, we don't take into account gaseous emission or heat transfer. Moreover, in order to obtain reproducible experimental conditions, we have used a fuel bed composed by Excelsior, further tests should be performed using shrubs as fuel bed.

3. Minor comments

(1) The abstract needs to be rewritten. Specific results and conclusions should be added.

Reply: The abstract has been rewritten.

(2) Section 2.1. A picture of samples needs to be added

Reply: As mentioned previously, a picture highlighting the different structure of samples has been added (figure 1).

(3) It's better to move Fig.1 to the beginning of section 2.3.1

Reply: As recommended, the figure 1 has been moved.

(4) Page 4, lines 21-23. It is not clear where the thermocouples were located, at the end of the tube or mid-height of the fuel brunch?

Reply: There are 2 tubes very close. One allow to place the plant, the other accommodate a thermocouple. Each tube allow an adjustment of elements in height. Thus, it is ensured that the thermocouple is positioned at mid-height of the branch. This part of the text has been rewritten.

(5) There is no reference to Fig. 3 in the text.

Reply: The figure is now cited in the text.

(6) Page 6, line 16. Temperature units need to be changed to K.

Reply: This mistake has been corrected.

(7) Figure 4. Axis ticks are needed on temperature axis. It is also hard to see them on other axes.

Reply: Axis ticks have been added and the thickness of axis has been increased for a better visualization of the graphics.
(8) Figure 5, Table 1-3. It would be worthwhile adding confidence levels.

Reply: As suggest by the reviewer, confidence levels have added in the table 1.

(9) I would recommend merging Fig. 5 and 6.

Reply: We have tried to merge these 2 figures but the rendering is not clear, we think that it is better to let the 2 figures separate for more visibility. If the reviewer wishes we can merge the 2 elements of each figure namely the temperature and the mass-loss but for a single specie and we can propose 3 small graphs as we did for the figure 5. However this disposition does not facilitate the comparisons between species.

(10) Table 3. A column with relative or absolute errors needs to be added.

Reply: The relative errors calculated from experimental results have be added in the table.

The new figures and the revised text have been joined to the answers to the reviewer n°1

---

## Author Response (AR2)

**Answers to reviewer 2**

I agree with another Reviewer, that edge effect could influence on the fire behaviour next to mass-loss prototype. I think discussion on this subject should be included in the paper.

*Reply: We also agree with the reviewer on the potential edge effect. Nevertheless, information elements on the subject have been included twice in manuscript. First, when describing the design of the prototype, we have mentioned that the device was sized to be one-fifth the width of the plot to burn. This ensures that the fire completely encompasses the system during its propagation.*
*We also talk about edge effect when we have defined the position of the device within the fuel bed. We have selected to place the prototype near the end of the fuel bed to avoid any edge effect and to ensure a steady state fire propagation.*

I believe the authors chose an incorrect way to present the influence of selected parameters on the mass loss rate in Figure 11. They are analysing the effect of the parameters on the mass loss rate and the mass loss rate is depicted on the diagram as a parameter. I would recommend that the authors change this Figure.

*Reply: The Kiviat diagram or spider web diagram was used here to represent on a two-dimensional plane five sets of multivariate data. Each axis, which starts from the same point, represents a quantified characteristic. We have included the mass loss to facilitate a detailed analysis (and make the ranking easier) of the elements influencing this data.*
*As well as their general comparison of surfaces (or point by point), this type of diagram is useful if the axes are correctly standardized according to the importance given to each characteristic.*

**Minor comments**

(1) Figure 2. It requires a note that the dimension of all the quantities is in cm or adding "cm" to each number.

*Reply: The information on the measurement unit has been add on the figure title.*

(2). Page 6, line 29. "", these previous works…". Which works? There is only one reference above (line 27).

*Reply: "These previous works" has been changed by "This previous work".*

(3). Page 6, line 31. A reference needs to be added.

*Reply: A reference has been added.*

(4). Page 7, line 9. There is a misprint. It should be "1 m" instead of "1".

*Reply: The unit has been added.*

(5). Page 7, line 13. "15 W/cm2" needs to be deleted.

*Reply: This part has been deleted.*

(6). Page 7, line 31. Figure 4 confuses the reader, as you said "The mass-loss prototype was placed near the end of the bed to ensure the steady state of the fire propagation." Please add the picture with the prototype at the end of the plot.

*Reply: Unfortunately, we don't have taken pictures in the final experimental conditions i.e. near the end of the plot. However, we can provide a picture when the prototype has been placed at the completely end of the plot. So, this picture has been added to the previous one, and for more clarity we have specified in the text and in the title of the figure that we present 2 tests with various configuration.*

(7). Authors used the different ways to express dimensions, e.g. m.s-1 or m/s. One way of writing should be used.
*Reply: The writing of the units has been standardized.*

(8). Figure 6. Was it an average temperature for each species? If yes, a note should be added.
*Reply: Actually, this is an average, this information has been added in the manuscript.*

(9). Figures 7 and 8 duplicate each other. I would recommend using only one Figure in the paper.
*Reply : We do agree with the reviewer, from our point of view only the figure 8 is relevant to highlight the difference between each specie. Nevertheless, in the previous review step, one of the reviewer has asked us to add the 3 repetition for each specie.*

(10). Figure 9. Was it an average heat flux? If yes, a note should be added.
*Reply: This is an average, this information has been added in the manuscript.*

(11). Table 4. A relative error (or other) between the experiment and simulation needs to be added.
*Reply: The relative error has been added in the table 4.*